# Effect of Top-Coat Thickness and Interface Fluctuation on the Residual Stress in APS-TBCs

**Weiling Zhao** [1,2,†]**, Zhongchao Hu** [3,†]**, Liang Wang** [1,2,*]**, Xintong Wang** [3]**, Qihao Wu** [3] **and Runpin Liu** [3]

1   State Key Laboratory of High-Performance Ceramics and Superfine Microstructure, Shanghai Institute of Ceramics, Chinese Academy of Sciences, Shanghai 201899, China; wlzhao_siccas@126.com
2   College of Materials Science and Optoelectronic Techonlogy, University of Chinese Academy of Sciences, Beijing 100049, China
3   Sany Heavy Industry Co., Ltd., Changsha 410100, China; zchu_siccas@126.com (Z.H.); wxt1160613349@163.com (X.W.); wuqh5@sany.com.cn (Q.W.); 15022287912@163.com (R.L.)
*   Correspondence: l.wang@mail.sic.ac.cn
†   These authors contributed equally to this work and should be considered co-first authors.

**Abstract:** This study focused on the numerical simulation of the distribution of residual stress in yttria-stabilized zirconia (YSZ) coatings prepared with atmospheric plasma spraying (APS). We particularly investigated the stress distribution around the interface between the top coat and bond coat. During thermal spray deposition, droplets and particles deposit on the substrate in a complex manner, causing interface waviness and non-uniform stress distribution. Therefore, residual stress is an important consideration when preparing thermal barrier coatings (TBCs). Residual stresses directly affect the performance of bond coats (BCs) and ceramic top coats (TCs). To accurately evaluate residual stress, we considered interface waviness and the thickness of the ceramic top coat and conducted a detailed analysis of stress distribution. The results show that compressive stress exists at the TC/BC interface, which may be caused by the mismatch in the thermal expansion coefficient between the YSZ top coat and the substrate, potentially leading to coating delamination. Moreover, the residual stress at the TC/BC interface significantly increases with an increasing YSZ thickness. When the top-coat thickness exceeds 300 μm, stress concentration and failure of the coating are likely to occur. Meanwhile, the optimized thermal spray experiment results confirm that the residual stress at the BC/YSZ interface of the thermal barrier coating is tensile stress, with a maximum value of 155 MPa, which is consistent with the finite element calculation results. Furthermore, the failure modes of TBCs with rough interface conditions are discussed in detail. Our research provides important guidance for TBC design and optimizing their performance.

**Keywords:** thermal barrier coatings; atmospheric plasma spraying; finite element modeling; residual stress; failure modes



## 1. Introduction

Thermal barrier coatings (TBCs) have extensive applications in modern aviation, aerospace, nuclear industry and other high-temperature structural components [1–3]. A YSZ top coat can effectively insulate heat and effectively reduce oxidation. Simultaneously, it can reduce large thermal stress and improve the service lifetime of a turbine. The droplets and particles are deposited onto the substrate using atomization, flight, collision, solidification and shrinkage. A high-temperature gradient will appear during thermal spraying, and the difference in thermal physical properties among the adjacent layers will further promote the development of residual stress in the coatings. Residual stress was induced in the coatings [4]. The residual stress in an as-sprayed coating exists inside the coating or at the interface between adjacent layers and will be superimposed into the spatial distribution of stress in the subsequent service process of the coating, which has a significant influence on the service performance of the coating when it is exposed to exterior conditions.

If residual compressive stress exists on the surface of the coating, the improvement in the fatigue strength in the coating is favorable, while the residual tensile stress that exists on the surface is not conducive to an improvement in the fatigue strength in the coating. The stress state has a significant influence on the subsequent service of the coating, including high and low-temperature cycles, long-term creep and low-cycle fatigue. However, an evaluation of the residual stress state of TBCs was significant, which was the premise for studying the delamination of coating and peeling. The complicated preparation process makes the residual stress state of TBCs very complex, especially the residual stress state near the interface [4]. It has a great influence on the service lifetime and failure behavior of coating at high temperature [5,6]. Therefore, the prediction and control of residual stress play a vital role in improving coating quality and prolonging its service lifetime. In the case of different spraying processes and coating thicknesses, the magnitude and distribution of residual stress were found to be directly influenced by the properties of the material when it was found in coatings. For decades, efforts have been made to understand and predict the residual stresses generated in multilayer film structures. Stoney et al. proposed the earliest description of stress [7]. However, TBCs were a typical porous multi-layer structure, and the stress–tensor relationship was nonlinear, which led to complicated mechanical properties. Experimental methods were used to evaluate the lifetime of TBCs, and the evaluation of interfacial stress was confirmed to be effective in the coating. Generally, the combination of X-ray diffraction, fluorescence spectrometry, the curvature method, etc. with computer simulation can improve the accuracy of calculating residual stress. The combination of non-destructive detection technology (Table 1) and computer simulation to calculate the residual stress of TBCs [7–11] will provide a technical possibility to verify the exact effect of coating geometric parameters on residual stress. In order to analyze the influence of various parameters on residual stress in TBCs, the thermal spray process can be regarded as the dynamic behavior of layer by layer depo-sition, and the corresponding model was used in the current modeling and simula-tion study. Specifically, the life and death element technology combined with the finite element simulation was used in this work [11–13] (Figure 1). The influence of YSZ coating thickness and rough interface on the distribution of residual stress between the bond coat and YSZ top-coat material is discussed when the bond-coat layer and YSZ top coat were sprayed on the substrate in turn.

**Table 1.** Non-destructive detection method [7–11].

| Measuring Method | Advantage | Disadvantage |
|---|---|---|
| Hole-drilling method | Reflects stress gradient changes | Coating sample is damaged |
| Curvature method | Real time detection of dynamic change in stress | A large deformation is required |
| X-ray stress diffractometer method | Convenient and quick analysis of the surface stress state | Measuring surface residual stress |
| Neutron diffraction | Depth of penetration (more than a few centimeters) | Strict equipment requirements |
| Raman spectroscopy | Measurement of surface stress and thickness direction stress | Difficult to measure thick coatings |

As shown in Figure 1, the initial substrate temperature of the model was t = 475 °C and the free air temperature was t = 25 °C.

As the model undergoes air cooling, the maximum convective coefficient reaches h = 8 W/(m$^2$·K).

During the spraying process, the convective heat transfer coefficient between the top of the model and the air is h = 100 W/(m$^2$·K).

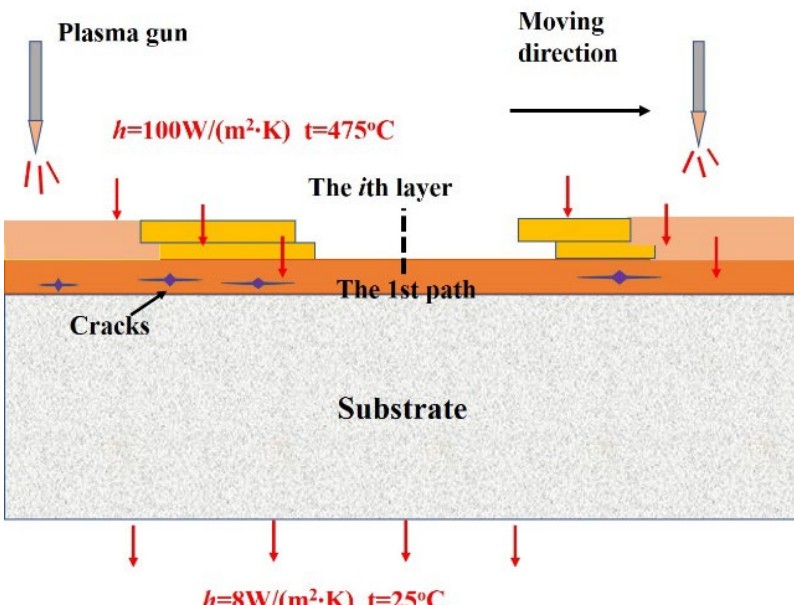

**Figure 1.** Model showing the deposition process of coatings.

## 2. Simulation Methods and Procedures

### 2.1. FE Model

The distribution of the temperature field and stress field in the as-sprayed TBCs was simulated in this paper using the Ansys APDL module. In the current study, the following assumptions were considered: (1) the BC layers and top coat were homogeneous and isotropic. (2) The insulated convection and air existed on the upper surface of the coating. (3) The creep deformation of each layer can be ignored. Creep generally occurs above 600 °C, and the service time needs to be long enough. The duration time of the coating from high temperature (stop feeding powder) cooling to the ambient temperature is short, usually less than an hour, and the whole coating system can be cooled from high temperature to room temperature. The short time and surface temperature of the as-sprayed coating is generally about 475 °C and the temperature is much lower than the creep temperature, that is, the creep effect has nearly no effect on the coating stress. Therefore, the creep effect was not considered in our current modeling and simulation process.

### 2.2. Boundary Conditions

To calculate the residual stress, Saint-Venant's principle was considered [14]. Figure 2 shows that the *n* layer of coating with a single thickness of $h_i$ is related to a substrate with a thickness of $h_s$, and $h_{total}$ represents the all thickness, which is calculated using [14]

$$h_{total} = h_S + \sum_{i=1}^{n} h_i \tag{1}$$

where the number of coatings is represented by *i* (1 to *n*), where 1 is directly attached to the substrate. Moreover, *k* represents the number of interfaces between the adjacent layers. The r-axis is collinear with interface *i*. Thus, interface *k* (*k* = *i* + 1), between layers *i* and *i* + 1, is located at $z = t_i$, and $z = t_n$ and $z = -t_s$ represent the free surfaces of the coating and substrate, respectively.

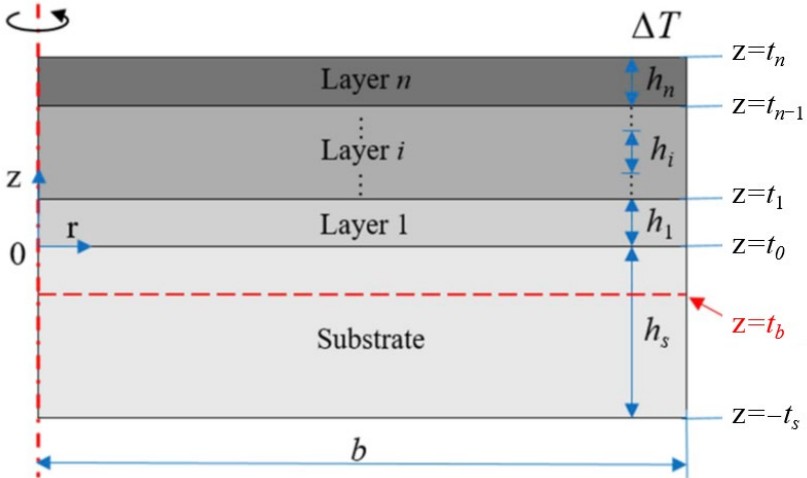

**Figure 2.** A two-dimensional axisymmetric multilayered model for the analytical calculation [14].

Hsueh's theory was taken into account in the current study [15], where $\Delta T$ represents the uniform temperature, and the radial stresses of the substrate $(\sigma_s^r)$ and the coatings $\sigma_i^r$ are respectively defined using

$$\sigma_s^r = \frac{E_s}{1 - v_s}\left(\frac{z - t_b}{\rho} + c - \alpha_s \Delta T\right) \tag{2}$$

$$\sigma_s^r = \frac{E_i}{1 - v_i}\left(\frac{z - t_b}{\rho} + c - \alpha_i \Delta T\right) \tag{3}$$

where $c$ is the uniform strain component. The bending axis is represented by $z = t_b$, where the value of the bending strain component is set as 0 (Figure 2). $c$, $t_b$ and $\frac{1}{\rho}$, are defined by

$$c = \frac{\frac{E_s h_s \alpha_s \Delta T}{1 - v_s} + \sum_{i=1}^{n} \frac{E_i h_i \alpha_i \Delta T}{1 - v_i}}{\frac{E_s h_s}{1 - v_s} + \sum_{i=1}^{n} \frac{E_i h_i}{1 - v_i}} \tag{4}$$

$$t_b = \frac{\frac{E_s h_s^2}{1 - v_s} + \sum_{i=1}^{n} \frac{E_i h_i}{1 - v_i}(2t_{i-1} + h_i)}{2\left[\frac{E_s h_s}{1 - v_s} + \frac{E_i h_i}{1 - v_i}\right]} \tag{5}$$

$$\frac{1}{\rho} = \frac{-6\left[\frac{E_s h_s \alpha_s \Delta T}{1 - v_s}\left(\frac{h_i}{2} + t_b\right) - \sum_{i=1}^{n} \frac{E_i h_i \alpha_i \Delta T}{1 - v_i}\left(t_{i-1} + \frac{h_i}{2} - t_b\right)\right]}{\frac{E_s h_s^2}{1 - v_s}(2h_s + 3t_b) + \sum_{i=1}^{n} \frac{E_i h_i}{1 - v_i}\left[6t_{i-1}^2 + 6t_{i-1}h_i + 2h_i^2 - 3t_b(2t_{i-1} + h_i)\right]} \tag{6}$$

when $i = 1$, the value of $t_{i-1}$ is set as 0. The distribution of residual stress along the BC/YSZ interface can be calculated using Equations (1)–(6).

In this study, the thickness of the bond coat and the 8YSZ top coat were set as 120 μm and 150~350 μm, respectively (Figure 3). Figure 3b shows that the left boundary was set using symmetric boundary conditions to restrict displacement in the x-direction, and the bottom boundary restricts the displacement in the y-direction. The material properties of the 8YSZ top-coat, NiCoCrAlY bond-coat and substrate as functions of temperature are listed in Tables 2–4 [11,16–21], respectively. In the calculation, the initial temperature of the substrate was assumed to be at room temperature, the process of coating droplets impacted and spread on the substrate at the melting point temperature, and the first step of the transient solution was carried out. According to the sequence of layer deposition, each thin layer was continuously "activated". The simulation time for each thin layer was set as 5 s, and the thickness of each layer was set as 20 μm (Figure 3). At the end of spraying, the result of the transient solution between the coating and substrate was taken as the initial condition.

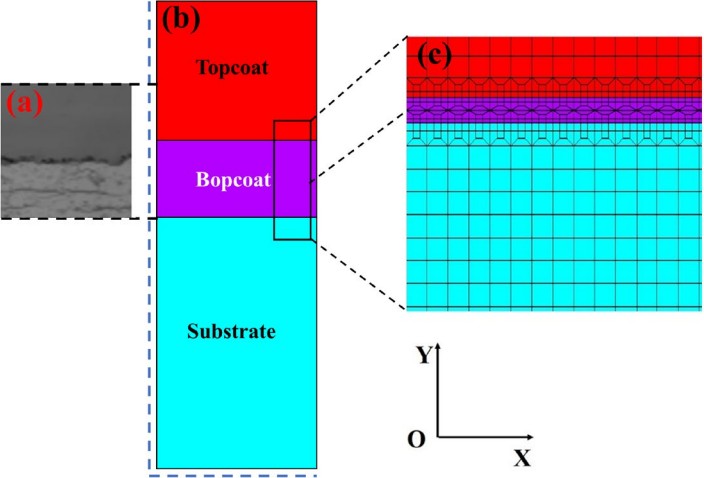

**Figure 3.** The FEM model of TBC. (**a**) SEM image of the TBC. (**b**) TBC system. (**c**) Refined mesh at the bond coat/top coat interface.

**Table 2.** Material properties of NiCoCrAlY for the finite element (FE) analysis [11,21].

| $T/^{\circ}$C | 20 | 200 | 500 | 700 | 1100 |
|---|---|---|---|---|---|
| $E$(GPa) | 152.4 | 143.3 | 136.7 | 126.4 | 41.3 |
| $\nu$ | 0.1 | 0.1 | 0.11 | 0.11 | 0.12 |
| $\alpha(10^{-6}\ ^{\circ}$C$^{-1})$ | 12.3 | 13.2 | 14.7 | 15.9 | 17.7 |
| $k$(W/m·K) | 4.3 | 5.2 | 6.4 | 8.6 | 10.2 |
| $c$(J/kg·$^{\circ}$C) | 501 | 547 | 598 | 638 | 781 |
| $\rho$(kg/m$^3$) | $5.28 \times 10^3$ | $5.28 \times 10^3$ | $5.28 \times 10^3$ | $5.28 \times 10^3$ | $5.28 \times 10^3$ |

**Table 3.** Material properties of substrate (GH4169) for the finite element (FE) analysis [19,20].

| $T/^{\circ}$C | 20 | 200 | 400 | 600 | 800 | 900 | 1100 |
|---|---|---|---|---|---|---|---|
| $E$(GPa) | 220 | 210 | 190 | 170 | 155 | 140 | 130 |
| $\nu$ | 0.31 | 0.32 | 0.33 | 0.33 | 0.33 | 0.34 | 0.35 |
| $\alpha(10^{-6}\ ^{\circ}$C$^{-1})$ | 14.8 | 15.2 | 15.6 | 16.2 | 16.9 | 11 | 17.5 |
| $k$(W/m·K) | 4.3 | 5.2 | 6.4 | 8.6 | 10.2 | 16.1 | 16.9 |
| $c$(J/kg·$^{\circ}$C) | 658 | 667 | 680 | 690 | 696 | 703 | 716 |
| $\rho$(kg/m$^3$) | $8.15 \times 10^3$ | $8.15 \times 10^3$ | $8.15 \times 10^3$ | $8.15 \times 10^3$ | $8.15 \times 10^3$ | $8.15 \times 10^3$ | $8.15 \times 10^3$ |

**Table 4.** Material properties of 8YSZ for finite element (FE) analysis [16–18].

| $T/^{\circ}$C | 20 | 200 | 500 | 700 | 1100 | 1200 |
|---|---|---|---|---|---|---|
| $E$(GPa) | 48 | 47 | 43 | 39 | 25 | 22 |
| $\nu$ | 0.1 | 0.1 | 0.1 | 0.1 | 0.1 | 0.1 |
| $\alpha(10^{-6}\ ^{\circ}$C$^{-1})$ | 10.4 | 10.5 | 10.7 | 10.8 | 10.9 | 11 |
| $k$(W/m·K) | 1.8 | 1.76 | 1.75 | 1.72 | 1.695.38 | 1.67 |
| $c$(J/kg·$^{\circ}$C) | 640 | 640 | 640 | 640 | 640 | 640 |
| $\rho$(kg/m$^3$) | $5.28 \times 10^3$ | $5.28 \times 10^3$ | $5.28 \times 10^3$ | $5.28 \times 10^3$ | $5.28 \times 10^3$ | $5.28 \times 10^3$ |

In the current study, the initial and reference temperatures of the substrate was 475 $^{\circ}$C, the bond coat was 1680 $^{\circ}$C, and the 8YSZ coatings was 2600 $^{\circ}$C, respectively. The convective heat transfer coefficient of 100 W/(m$^2$·K) was applied around the sample, and the natural cooling time was set as 3600 s [11,22]. The final residual stress results were obtained at the YSZ coating/BC interface.

## 3. Results and Discussion

To study the residual stress numerically, the edge effect was considered [23–25]. The performance of the TBCs was decreased because of the distribution of stress concentration,

which promoted crack initiation in the coating. In the zone where large stress concentration can be seen at the right edge of the TBCs, due to the different CTE among the adjacent layers, the appearance of larger stress will lead to the delamination of coatings, especially near the YSZ/BC interface.

Although the bond coat had a positive effect on decreasing the residual stress of TBCs, it was concluded that the concentration of residual stress may exist along the YSZ/BC interface when the position was close to the right of TBCs, as shown in Figure 4a–c. We observed the phenomenon that the maximum tensile stress was concentrated in the main zone, which tended to exist along the top coat/bond coat interface. This would cause crack initiation along the interface of YSZ/BC. Moreover, the crack initiation, propagation and coalescence of cracks were produced due to the larger tensile stress existing in the YSZ layer [26–28]. However, the interfacial cracks were generally caused by shear stress ($\tau_{xy}$), which would lead to the spallation of the interface layer [29]. $\tau_{xy}$ is a key factor influencing the failure of TBCs. Micro-cracks commonly occur in most of the YSZ/BC interface, leading to complex interrelationships among the layers and accelerating coating spalling. Therefore, to understand the failure mechanism of TBCs, the stress distribution along the TC/BC interface was systematically analyzed.

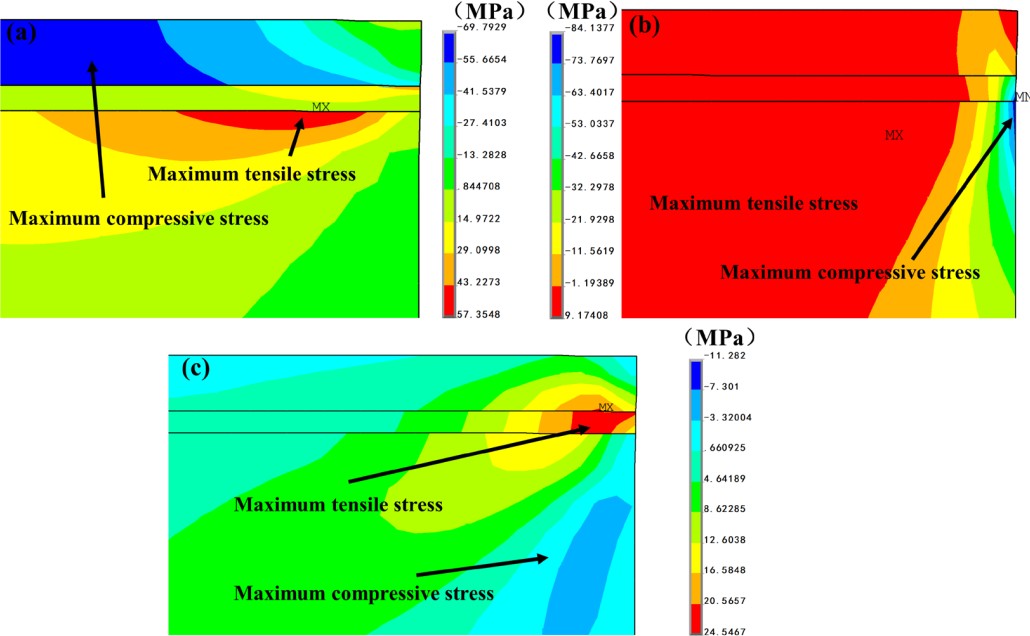

**Figure 4.** Stress distribution in TBCs: (**a**) radial stress, (**b**) axial stress and (**c**) shear stress.

### 3.1. Effect of YSZ Thickness on Residual Stress in the Coating

Thickness was an important factor for the TBCs in this study. The top layer was set to 150~350 μm, and the interfacial residual stress along the YSZ/BC interface was analyzed. Most of the radial stress was compressive stress, which can be observed in the coating. Furthermore, this would lead to a large stress concentration existing at the YSZ/BC interface. Additionally, the concentration of residual stress tends to exist at the YSZ/BC interface, which can be seen in Figure 5a–c. The influence of the thickness of 8YSZ affected the interfacial residual stress at the interface. Combined with Figure 4, it can be seen that a serious stress concentration existed near the edge of the interface with the increasing YSZ thickness. As the coating became thicker, the axial compressive stress, radial stress and tangential stress gradually developed, which was because the redistribution of shear stress tended to exist at the interface between the YSZ coating and substrate when the coating was cooled. The delamination of the TBCs was influenced by large tensile stress along the YSZ/BC interface [30]. In fact, the thickness of TBCs was a significant factor in evaluating the value of residual stress. Therefore, we mainly studied the influence of thickness on the

distribution of residual stress along the YSZ/BC interface. In addition, the distribution of residual stress of interfacial stress was connected with the distribution of the temperature field, and during the subsequent high-temperature service condition, the bond coat lost its stability. This will lead to a complicated residual stress state and has an important influence on the crack generation of TBCs for practical applications. Many studies have demonstrated that thermal stress and growth of thermally grown oxide will drive the complicated interaction between a microcrack and an interface, and the initiation and propagation of cracks were formed at the BC/YSZ interface of TBCs [31–35]. To calculate the residual stress, the distribution of the temperature field was considered [11], and the residual stress often accelerated the production of cracks in TBCs. This will affect the failure modes and eventually reduce the lifetime of TBCs.

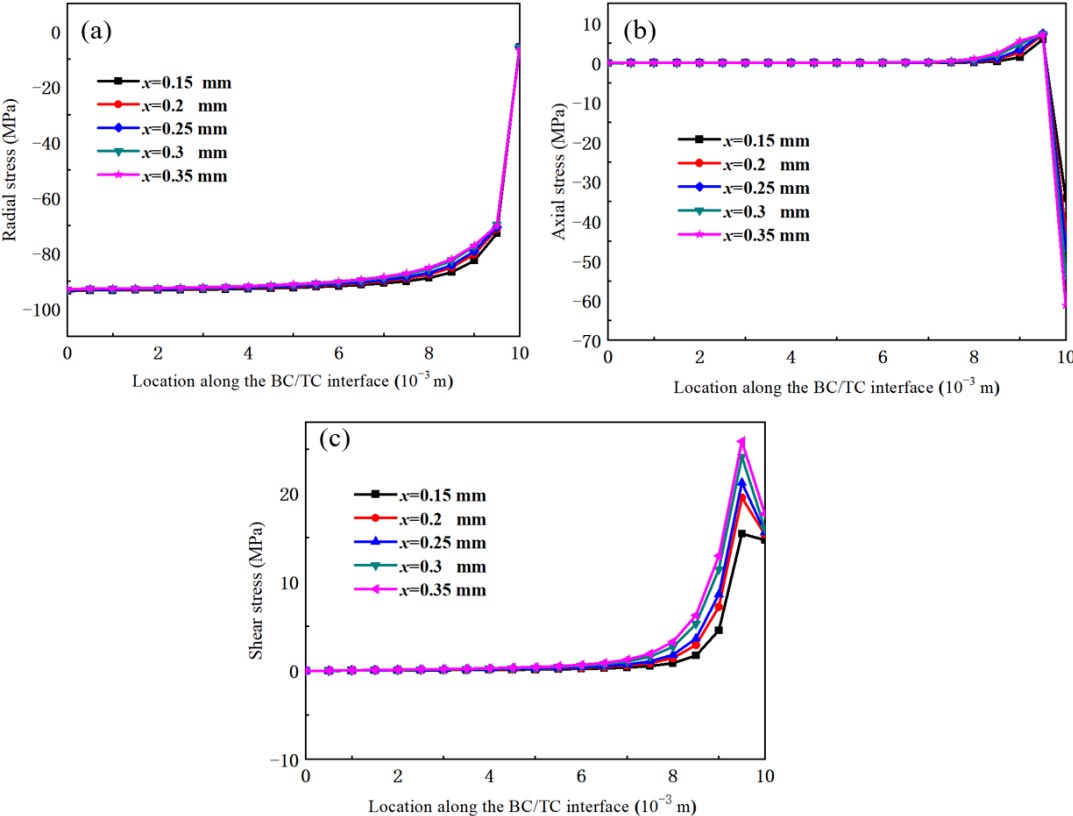

**Figure 5.** Stress distribution curve along the BC/TC interface: (**a**) radial stress, (**b**) axial stress and (**c**) shear stress in TBCs.

With an increase in the coating thickness, the stress concentration of $\tau_{xy}$ suddenly decreased, which existed at the edge of the YSZ/BC interface. Further, crack initiation occurred. In fact, the thickness of the YSZ coating was thin and did not play a substantial role in improving the effect of insulation. However, as the YSZ coating was thick, it was inevitable that large residual stress was generated during spraying. When the thickness of the YSZ coating was restricted by a certain value, the actual position of the residual stress did not change due to the change in thickness. The spalling of coatings occurred due to the appearance of large stress when the TBCs were cooled after thermal spraying. Meanwhile, it can restrain the spalling or flaking of TBCs, delay the delamination of the coating and improve the durability of TBCs. The results demonstrated that the shear stress suddenly increases when the thickness of YSZ exceeds 300 µm. Therefore, the thickness of YSZ coating should be selected as 300 µm, as this thickness will greatly reduce the residual stress in the whole TBCs.

*3.2. Influence of Interface Fluctuation on Residual Stresses*

To obtain more accurate calculation results of residual stress, the stress distributions along the BC/YSZ interface were considered due to the coating's tendency to crack along the interface of BC layer/YSZ coatings. In addition, the interaction between internal factors and external factors led to a complicated stress distribution at the position of the interface. To discuss the existence of stress distribution, the interface fluctuation in YSZ coatings was discussed to investigate the residual stress at the BC/YSZ interface. Figure 6 indicates that the stress distribution in TBCs with a cosine wave interface ($A$ = 5 μm, $\lambda$ = 20 μm) was taken into account. Figure 6a indicates that the radial stress of YSZ coatings was compressive stress, and the result shows that the value of the maximum compressive stress is −124.8 MPa at the peak. In addition, the appearance of maximum tensile stress is 134.4 MPa, existing at the valley. Further, some micro-cracks were produced along the YSZ/BC interface due to the interaction between the complicated interfacial stress and interface fluctuation, eventually causing the spalling of layers. Figure 6b demonstrates that the tensile stress concentration tended to occur near the position of the peak in the YSZ/BC interface, and the value of the maximum tensile stress is 151.1 MPa. Furthermore, the maximum compressive stress is 260.7 MPa, and the appearance of stress concentration was observed at the valley of the YSZ/BC interface. It would be useful to restrain stress concentration for the microcracks that tend to exist toward the BC/YSZ interface. Figure 6c shows an analysis of the distributions of shear stress, and the results indicate that the distributions of tensile stress were observed when it existed at the peak and valley, and compressive stress occurred at the crest and valley. The result shows that the concentration of compressive stress seemed to be toward the 8YSZ from peak to valley, and it was found that the tensile stress concentration existed in the bond coat from crest to valley. Subsequently, the sudden change took place om the shear stress when it occurred at the peak and the valley of TBCs.

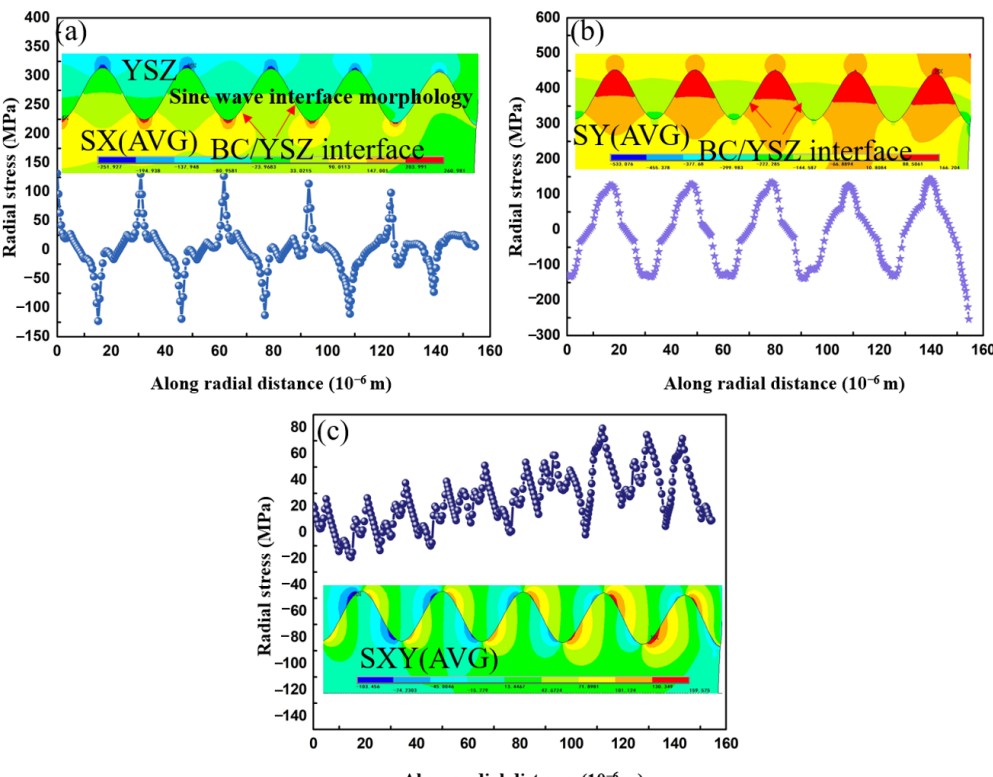

**Figure 6.** Stress change along the radial direction at the BC/YSZ interface: (**a**) radial stress, (**b**) axial stress and (**c**) shear stress in TBCs.

The edge of the BC/YSZ interface was influenced by the stress concentration of TBCs, which influenced the bonding strength and reduced the stability of coatings. Moreover, the radial compressive stress was large along the BC/YSZ interface, and the peeling of the coating was affected by the axial tensile stress. Additionally, it was found that the interaction between different wavelengths and amplitude had a strong influence on the behavior of the stress state. Consequently, Figure 7 indicates that to determine the interfacial stress of TBC ($A$ = 3~7, $\lambda$ = 20 µm), the conditions of five kinds of amplitude were taken into account.

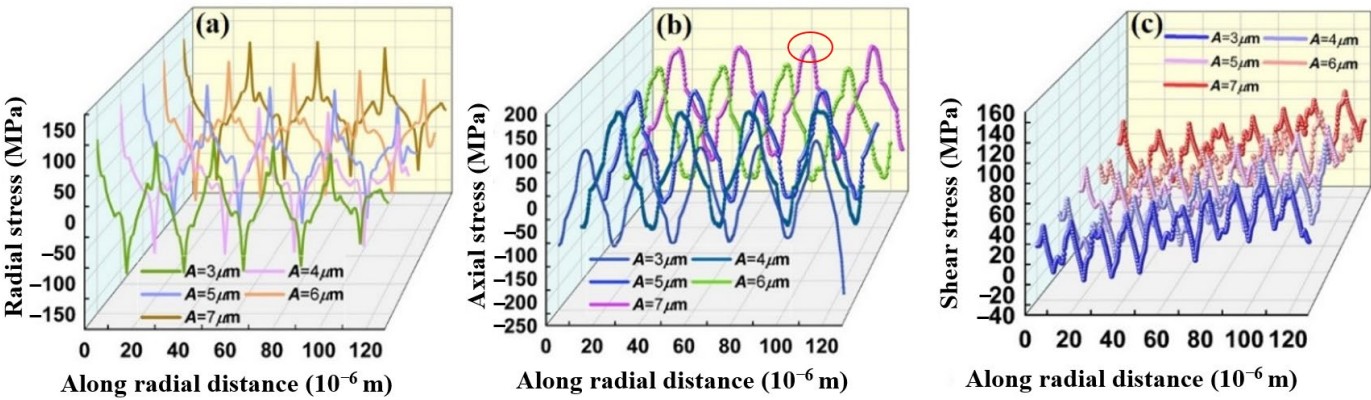

**Figure 7.** Stress changes along the radial direction at the BC/YSZ interface with different amplitudes: (**a**) radial stress, (**b**) axial stress and (**c**) shear stress. The wavelength ($\lambda$) is equal to 20 µm.

The tensile stress gradually changed when the amplitude increased. The amplitude was associated with a smooth interface, and the lower the amplitude, the smoother is interface. It was found that the concentration of tensile stress appeared at the peak of the YSZ/bond coat interface. Therefore, the stability of a smooth interface would be affected. As shown in Figure 8, the interfacial stress of TBC ($A$ = 4, $\lambda$ = 12~20 µm) was considered with five kinds of wavelengths, and higher tensile stresses were reduced by the increase in wavelength, which tended to occur at the peak. Furthermore, the result indicates that the axial stress in TBCs was compressive stress. The compressive stress concentration occurred toward the valley, and the concentration of tensile stress existed at the crest along the YSZ/BC interface. This would make a crack initiate because of the larger tensile stress and accelerate the interfacial crack propagation to form a large crack length along the YSZ/BC interface. The residual stress would be effectively reduced, and the generation of a crack would be probability decreased at the YSZ/BC interface.

It is important to note that the plots presented in Figures 7 and 8 depict the interfacial stress behavior of TBC under different amplitude and wavelength conditions. The plots represent a superimposition of multiple amplitudes and wavelengths, chosen specifically for the analysis. By considering various amplitudes and wavelengths, we can gain insights into the influence of these parameters on the stress distribution and stability of the coatings. This approach allows us to observe the overall trends and interactions between different amplitude and wavelength combinations.

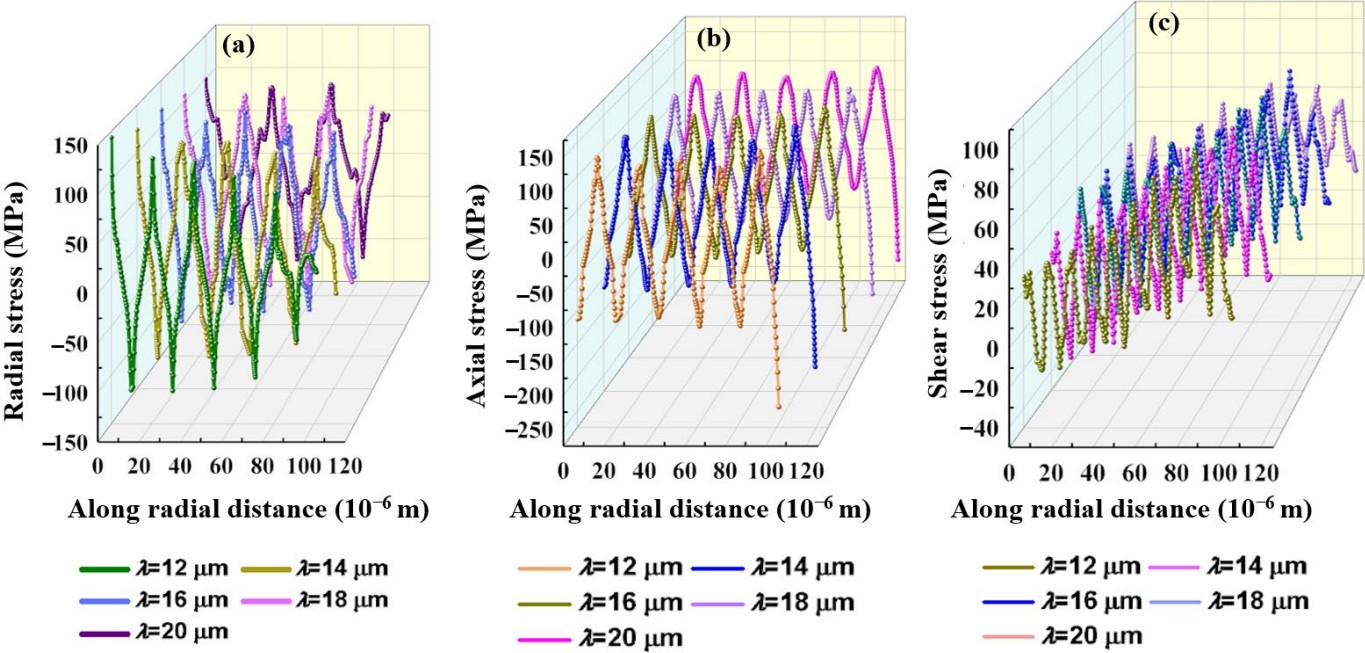

**Figure 8.** Stress changes along the radial direction at the BC/YSZ interface with different wavelengths ($\lambda$): (**a**) radial stress, (**b**) axial stress and (**c**) shear stress. The amplitude is equal to 4 µm.

## 4. Experiment

Figure 9 shows the macroscopic morphology of the surface of the TBCs. From Figure 9a, it can be observed that the thickness of the plasma-sprayed thermal barrier coating is 300 µm, and the surface is not dense enough, indicating considerable roughness. Some unmelted particles are also found in the form of splats. At the same time, a large number of pores and micro-cracks are evident, which are likely caused by residual stress accumulated during spraying. Figure 9b shows an enlarged view of the surface of the plasma-sprayed TBCs. The coating surface is composed of layered structures with fully melted powder particles that spread out into a layered form after plasma spraying. Due to the accumulation of high thermal stress during spraying, significant large residual stress is generated, which affects the surface morphology of the plasma-sprayed TBCs. Furthermore, numerous long cracks are observed on the surface area, primarily attributed to residual tensile stress on the surface, confirming the results of the simulation. Figure 9c illustrates the EDS distribution on the surface of the plasma-sprayed thermal barrier coating. It can be seen that the surface element contains 68.77% Zr, 25.63% O, and 5.6% Y. The content of $ZrO_2$ is 92.89%, while $Y_2O_3$ accounts for 7.11%.

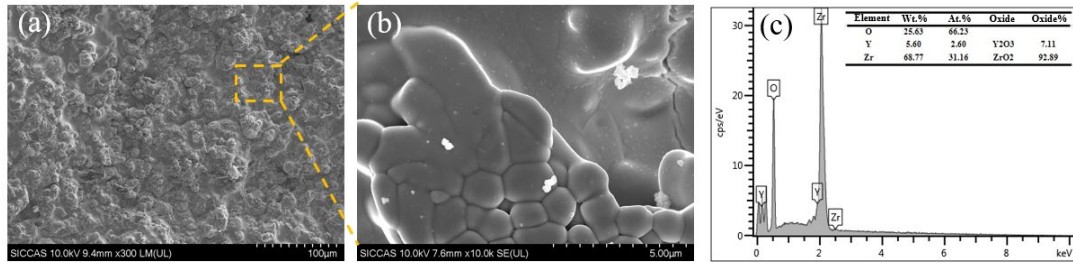

**Figure 9.** The surface morphology and EDS of TBCs. (**a**) YSZ. (**b**) Left box amplified. (**c**) EDS.

Figure 10a shows that the cross-sectional morphology of the nanostructured thermal barrier coatings bonded well with the substrate, and the interface between the ceramic layer and the bond coat was clearly visible. At the interface, a large number of cracks and pores were found, as shown in Figure 5b. The accumulation of residual stresses in this

zone during spraying resulted in excessive residual shear and tensile stresses, which led to crack initiation and propagation in this region. Meanwhile, at the interface between the ceramic layer and the bond coat, a sinusoidal interface was observed. The accumulation of tensile and compressive stresses from plasma spraying caused changes in the morphology of the thermal barrier coating interface, especially on the left side of the sinusoidal interface. Some interface sections showed small undulations, possibly due to the effect of tensile stress. The changes in interface morphology validated the finite element calculation results, as shown in Figure 6. Randomly distributed pores were also found in the cross-sectional morphology of the ceramic layer. The presence of pores has a certain heat dissipation effect. When the preparation is completed, the temperature field inside the coating can pass through the pores to reach the top of the ceramic layer, ensuring a stable internal heat distribution and improving the stability of the coating to a large extent. This better protects the substrate. On the other hand, an increase in pores can reduce the toughness of the coating. Under shear stress, cracks will initiate in this area and propagate along the interface under larger compressive stresses, greatly reducing the service life. At the same time, under the influence of residual tensile stress, interface cracks are prone to form vertical cracks, allowing oxygen to smoothly enter the interior of the bond coat and further accelerate the damage to the substrate. Figure 10b–f shows the EDS distribution maps of the cross-section of the plasma-sprayed TBCs. From these figures, it can be seen that oxygen (O) is distributed relatively evenly at the interface, as shown in Figure 10b. Figure 10c shows the distribution of aluminum (Al) at the interface, which is unevenly distributed with more Al in the bond coat and a small amount diffused into the ceramic layer. Zirconium (Zr) is mainly concentrated in the ceramic layer, as shown in Figure 10d, and does not appear in the bond coat. Chromium (Cr) and other elements are distributed relatively evenly in the bond coat, with a small amount diffusing to the interface between the ceramic layer and bond coat after spraying. When external oxygen enters, a small amount of $Cr_2O_3$ will form at the BC/TC interface (Figure 10e). Figure 10f shows the distribution of nickel (Ni) at the interface, which is relatively uniform in the bond coat.

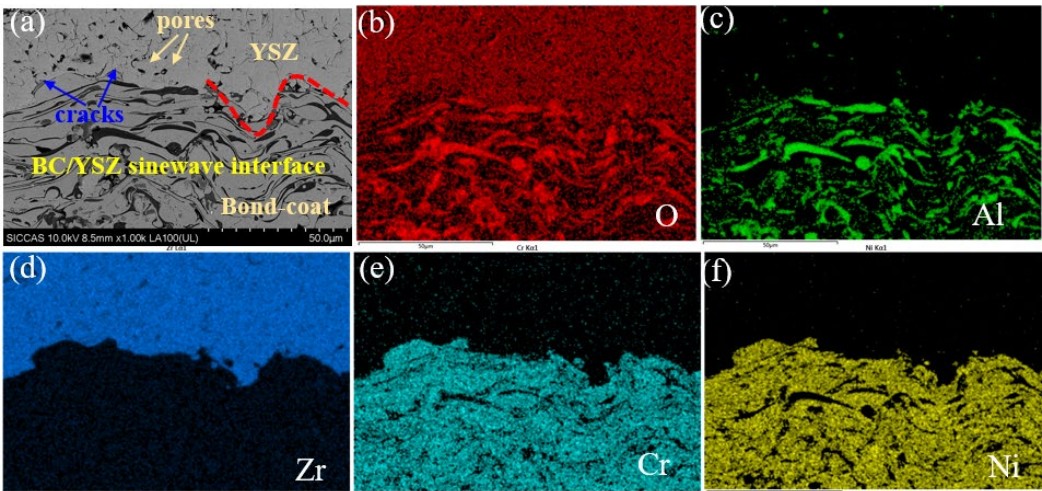

**Figure 10.** SEM of BC/YSZ interface: (**a**) cross-section of TBCs, (**b**) O, (**c**) Al, (**d**) Zr, (**e**) Cr and (**f**) Ni.

The surface stress result for the nanostructured 8YSZ obtained using the finite element calculation was −92 MPa, and the error between the measured value and theoretical value was 8.5%, which was in good agreement. The diffraction peaks on the surface of the thermal barrier coating changed with different diffraction angles, as shown in Figure 11. $\theta_0$ represents the initial diffraction angle when measuring the coating surface. The surface elastic modulus $E$ of the nano-YSZ coating was 63 GPa, and the Poisson's ratio $\nu$ was 0.3. The residual stress of the nano-YSZ coating was calculated using Formula (7), which was −84.2 MPa. The finite element calculation result for the surface stress of nano 8YSZ in

Figure 5b was −92 MPa, with an error between the measured value and theoretical value of 8.5%, which was in good agreement.

$$\sigma_\phi = \frac{E}{2(1+\nu)} \cot\theta_0 \frac{\pi}{180°} \frac{\partial 2\theta_{\varphi\phi}}{\partial \sin^2\varphi} \tag{7}$$

where $\theta_0$ represents the diffraction angle with no stress, $\phi$ is the angle between $\sigma_\phi$ and $\sigma_1$ and $\varphi$ is the angle between the normal vector of the diffracting plane and the normal vector of the sample surface. $E$ is the elastic modulus of the coating and $\nu$ is the Poisson's ratio. $\frac{\partial 2\theta_{\varphi\phi}}{\partial \sin^2\varphi}$ represents the slope corresponding to the maximum diffraction intensity.

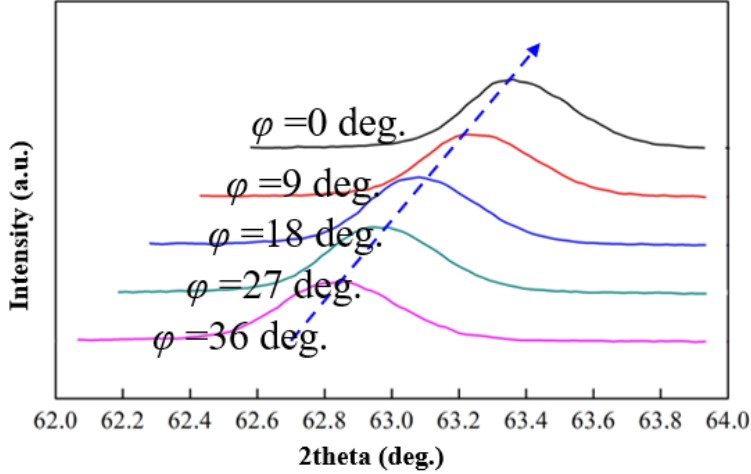

**Figure 11.** Diffraction peaks at different angles of TBCs surface.

The Renishaw in Via laser confocal Raman microscope with a wavelength of 785 nm from the Shanghai Institute of Ceramics, the Chinese Academy of Sciences, was used to test and analyze the BC/YSZ interface of the two samples. The scanning range was 100–3500 cm$^{-1}$. The residual stress of the YSZ coating showed that the offset of the Raman characteristic peak around 640 cm$^{-1}$ was proportional to the stress it received. There is a relationship between the residual stress and the Raman spectrum, which can be expressed as follows [36]:

$$\overline{\sigma_{xx}} = \overline{\sigma_{yy}} = \frac{\Delta\omega_0}{2\overline{\overline{\prod}}} \tag{8}$$

where $\Delta\omega_0$ represents the Raman shift and $\overline{\overline{\prod}}$ represents the stress coefficient.

Figure 12 shows the Raman spectra of the TBCs, where Figure 12a shows the YSZ spectrum curve with no stress, and Figure 12b shows the spectrum curve of nano-YSZ. It can be seen from Figure 12a that the Raman characteristic peak of the YSZ material with no stress is 634.2 cm$^{-1}$. In Figure 12b, the Raman characteristic peak of nano-YSZ shifts to the left, and the characteristic peak value is 633.7 cm$^{-1}$. The Raman characteristic peak of the stress-free ceramic layer is greater than that of the nano-coating, indicating that the residual stress at the BC/YSZ interface mainly manifests as tensile stress after spraying. The stress coefficient $\Pi$ of the YSZ coating is 1.61 cm$^{-1}$/GPa. The residual tensile stress at the BC/YSZ interface in the thermal barrier coating can be calculated as 155.2 MPa by substituting it into Equation (8). Compared with the theoretical calculation value of 141.3 MPa, the two values are almost the same, which confirms our calculation result. At the same time, we found that Wang Liang, from the Harbin Institute of Technology, used plasma spraying to prepare thermal barrier coatings and measured the residual stress distribution when the ceramic layer thickness was 300 μm. The maximum residual stress was 124.4 MPa [37]. There is some difference between their results and the results of this experiment, which is mainly due to differences in the process parameters for preparing the thermal barrier

coating. Therefore, it can be considered that the results of this experiment are in agreement with previous studies.

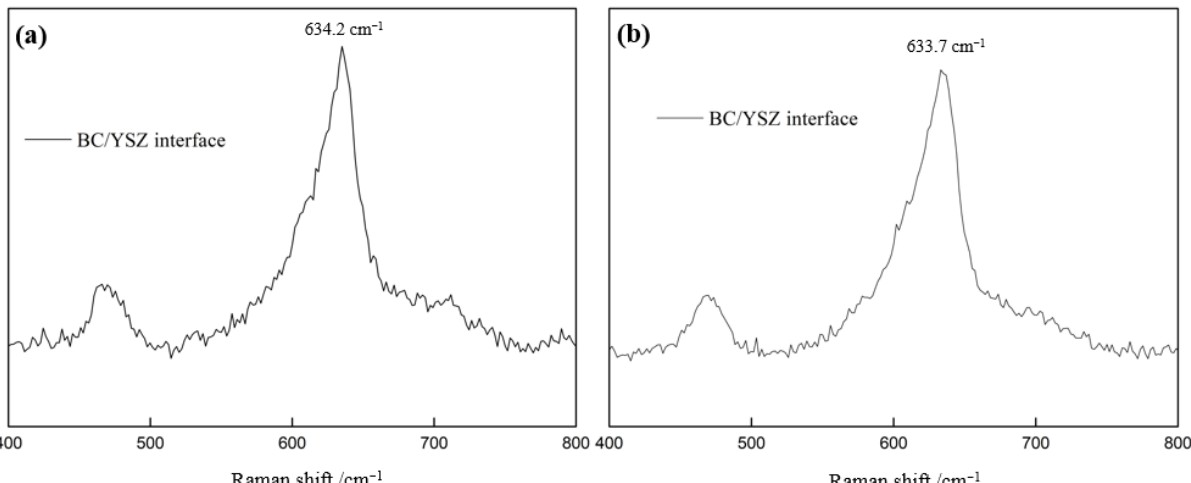

**Figure 12.** Raman diffraction peak of TBCs at the BC/YSZ interface: (**a**) reference peak and (**b**) nano-YSZ.

## 5. Possible Failure Modes of TBCs

The distribution of residual stress was extremely complicated when the YSZ coating was deposited onto the superalloy substrate, the ceramic powder was heated with a plasma flame, and the insufficient molten state, overlap behavior and residual stress made the as-sprayed coatings produce a large number of micro-cracks and pores. Here, the production of surface cracks of coatings was mostly related to the radial stress. The shear stress concentration usually tends to exist at the position of the peak of the YSZ/BC interface, resulting in complicated stress at the interface. Even stress will lead to the failure of coatings easily. Once large stress exceeds the fracture strength of TBCs, it will accelerate the spalling of coatings. Figure 13 shows the cracks that may occur inside the coating under different stress states. According to previous studies, it was found that large residual stress would cause the crack initiation and propagate along the interface between 8YSZ and the bond coat [11,19,22] (Figure 13a). If the residual tensile stress generated on the surface of the coating is greater than the bonding strength between the upper and lower or the front and rear adjacent splats, the cracks in the form shown in Figure 13a may occur when the adjacent splats begin to slip and escape under the action of the residual stress. Residual compressive stress existed at the YSZ/BC interface, which is beneficial for closing cracks [11]. It can further prevent crack propagation and slow down the spalling rate of the YSZ coating. However, the failure of TBCs was affected by the stress concentration of residual compressive stress; moreover, the failure modes of TBCs were related to the different residual stress states. It was found that the cracks and pores were distributed irregularly at the interior of the coatings, and a large residual compressive stress occurred, which affected the delamination of TBCs (Figure 13d). Considering the loading of cyclic residual stress, it is generally believed that the newly formed cracks will propagate toward to the 8YSZ/bond coat interface. The YSZ/BC interface would accelerate the eventual spallation of the coating and reduce the reliability of TBCs.

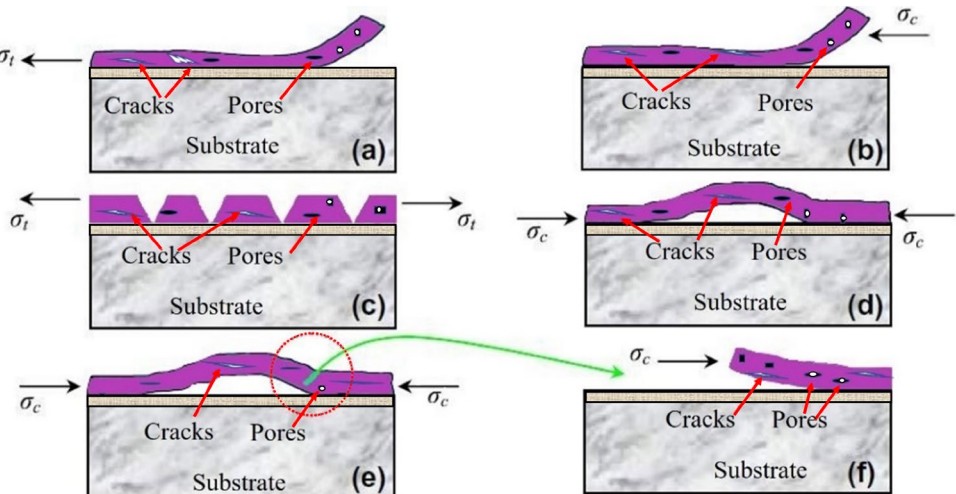

**Figure 13.** Failure models of TBCs under different states of residual stress: (**a**) the existence of large tensile stress, (**b**) failure at the edge of coating, (**c**) bridged crack, (**d**) bulking and (**e**) the delamination of the coating driven by bulking [35]. The crack initiation and propagation were driven by compressive stress at the YSZ/BC interface (**f**), the relative fracture strength of the YSZ/BC interface and the coating were the main factors in determining the spallation behavior of TBCs.

## 6. Conclusions

Based on the current finite element modeling, the thickness of 8YSZ was designed and optimized, and the residual stress distribution at the interface was analyzed using the "birth" and "death" element method. The following conclusions were obtained:

(1) The thickness of the YSZ coating significantly affects the residual stresses near the BC/YSZ interface. The radial stress at the BC/YSZ interface is primarily a compressive state, with a maximum compressive stress of −92 MPa. The axial stress is also a compressive state, while the shear stress is mainly a tensile state, with a maximum tensile stress of 28 MPa. In practical engineering applications, if the coating is too thick, it may result in significant residual stresses and make it prone to delamination after thermal spraying is finished. On the other hand, if the coating is too thin, it will reduce the thermal insulation effect. Based on the assumptions made in the current study, a coating thickness of 0.3 millimeters is considered to be the optimal thickness. However, it is important to emphasize that the optimal coating thickness is determined by multiple factors, including APS process parameters, coating materials and service conditions. Therefore, in actual engineering practice, it is necessary to comprehensively consider these factors. The ideal coating thickness should be determined using the finite element simulation combined with the experimental validation.

(2) A rough interface was used to simulate the coating deposition process, which was closer to the actual spraying process. Earlier crack initiation and propagation led to the formation of a larger crack, which was increased by higher amplitude and lower wavelength. This approach provides guidance to prepare TBCs with excellent performance.

(3) The optimized results were used to prepare a nano-thermal barrier coating. X-ray diffraction (XRD) measurements showed that the residual stress on the surface of the nano-thermal barrier coating was −84.2 MPa. Additionally, Raman spectroscopy was used to measure the residual stress at the interface between the bonding layer (BC) and the ceramic top coat (YSZ) of the nanothermal barrier coating. Experimental results indicated that there was predominantly tensile stress at the interface, with a maximum tensile stress of 155.2 MPa. These experimental findings are consistent with the simulation results.

(4) In this study, the finite element simulation is used to judge the stress state and stress magnitude in the thermal barrier coatings. This method can analyze the initiation and propagation of cracks in the coating and the eventual failure of the coating

and provide theoretical support for the preparation process optimization of thermal barrier coatings.

(5) In future work, the impact stress of particles will be incorporated into the current finite element model to better simulate the deposition process of thermal barrier coatings (TBCs) in plasma spraying. Additionally, the dynamic evolution of thermally grown oxide (TGO) will be considered to further investigate how internal cracks propagate near the TGO region.

**Author Contributions:** Conceptualization, W.Z. and Z.H.; methodology, Z.H.; software, W.Z.; validation, L.W. and X.W.; investigation, Q.W.; resources, R.L.; data curation, Z.H.; writing—original draft preparation, W.Z.; writing—review and editing, W.Z.; visualization, Z.H.; supervision, Z.H.; project administration, L.W.; funding acquisition, L.W. All authors have read and agreed to the published version of the manuscript.

**Funding:** This research was jointly supported by a sub-project of the Key Basic Research Projects of Basic Strengthening Program (Grant No. 2020-JCJQ-ZD-172-05), the National Defense Basic Research (JCKY2021603B007), the Training Program of the Major Research Plan of the National Natural Science Foundation of China (No. 91960107), financial support from the National Natural Science Foundation of China (Nos. 52375222, 51671208), and the National NSAF (Grant No. U1730139). This work was also supported by the Youth Innovation Promotion Association of the Chinese Academy of Sciences (Grant No. 2017295) and the Natural Science Foundation of Shanghai (No. 19ZR1479600).

**Institutional Review Board Statement:** Not applicable.

**Informed Consent Statement:** Not applicable.

**Data Availability Statement:** The data that support the findings of this study are available from the corresponding author, Liang Wang, upon reasonable request.

**Conflicts of Interest:** The authors declared that they have no conflict of interest.

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
