# Peer review of "Effect of Top-Coat Thickness and Interface Fluctuation on the Residual Stress in APS-TBCs"

_coatings, doi:10.3390/coatings13091659_

Round 1

Reviewer 1 Report

Effect of thickness of top-coat and interface fluctuation on residual stress of APS-TBCs – Z. Weiling et al.

General Comments: FE-analysis of YSZ coatings fabricated by plasma spraying is presented. Although, the work is interesting, the paper is very poorly written. Major revision is suggested before the manuscript can be accepted. My specific comments are provided below.

Specific Comments:

1.       Rewrite abstract, e.g. Line 17 doesn’t make any sense.

2.       Line 19: what is meant by interface fluctuation here?

3.       Rewrite Line 31.

4.       Provide reference for Figure 1.

5.       Table caption must read “ …. Finite Element (FE) analysis.”

6.       Line 142: “currant” must read “current”. It is very confusing to follow the draft and must be carefully revised.

7.       Was there any mesh sensitivity study conducted in terms of thru-thickness direction? How does the max./min. stress change according to mesh density?

8.       Carpet plots presented in Figures 7 & 8 look good. However, this is a mere superimposition of various amplitudes which must be clearly stated in the main body.  

9.       Residual stress, by definition occurs due to mismatch in thermal coefficients. Hence, (1) in conclusion must be removed.

10.   Clearly present the observed trends in conclusions and outline the scope for future work (if any) in conclusions.

Extensive editing of English language required.

Reviewer 2 Report

In this paper, the authors present a series of computer simulations using the finite element method to analyze the stress state and stress magnitude of thermal barrier coatings, the propagation of cracks in the coating, and the eventual failure of the coating used as a thermal barrier.

The paper presents interesting and relevant results that justify its publication after some corrections.

1 - Minor corrections:

a- A text explaining the values presented in Table 1 and the configuration of Figure 1, must be included after Figure 1;

b- Authors must indicate in the text which criteria are defined to determine the thickness of the coating (page 4, line 124).;

2- Major Corrections:

a- The numerical results presented require validation so that they are consistent. To validate the results, I recommend including a table comparing the results obtained with experimental results or results already available in the literature. A more detailed discussion of the comparisons should be included in the paper. After the inclusion of the table, the conclusions must be updated.

Round 2

Reviewer 1 Report

The revised version may be accepted in its current form. 

Author Response

Many thanks to you for your former comments.

Reviewer 2 Report

The suggestions were incorporated into the manuscript, in my opinion the article can be published.

Author Response

(The authors gave the same response as above.)
